# Are older adults of Rohingya community (Forcibly Displaced Myanmar Nationals or FDMNs) in Bangladesh fearful of COVID-19? Findings from a cross-sectional study

Sabuj Kanti Mistry[1,2,3]*, A. R. M. Mehrab Ali[1,4], Farhana Akther[5], Prince Peprah[2], Sompa Reza[6], Shaidatonnisha Prova[6], Uday Narayan Yadav[2]

1 ARCED Foundation, Dhaka, Bangladesh, 2 Centre for Primary Health Care and Equity, University of New South Wales, Sydney, Australia, 3 BRAC James P Grant School of Public Health, BRAC University, Dhaka, Bangladesh, 4 Innovations for Poverty Action, New Haven, Connecticut, United States of America, 5 Mawlana Bhashani Science and Technology University, Tangail, Bangladesh, 6 Society for Health Extension and Development (SHED), Cox's Bazar, Bangladesh

* smitra411@gmail.com

**Data Availability Statement:** All relevant data are within the manuscript and its Supporting information files.

## Abstract

### Aim

This study aimed to assess the fear of COVID-19 and its associates among older Rohingya (Forcibly Displaced Myanmar Nationals or FDMNs) in Bangladesh.

### Method

We conducted a cross-sectional survey among 416 older FDMNs aged 60 years and above living in camps of Cox's Bazar, Bangladesh. A semi-structured questionnaire was used to collect information on participants' socio-demographic and lifestyle characteristics, pre-existing non-communicable chronic conditions, and COVID-19 related information. Level of fear was measured using the seven-item Fear of COVID-19 Scale (FCV-19S) with the cumulative score ranged from 7 to 35. A multiple linear regression examined the factors associated with fear.

### Results

Among 416 participants aged 60 years or above, the mean fear score was 14.8 (range 8–28) and 88.9% of the participants had low fear score. Participants who were concerned about COVID-19 ($\beta$: 0.63, 95% CI: -0.26 to 1.53) and overwhelmed by COVID-19 ($\beta$: 3.54, 95% CI: 2.54 to 4.55) were significantly more likely to be fearful of COVID-19. Other factors significantly associated with higher level of fear were lesser frequency of communication during COVID-19, difficulty in obtaining food during COVID-19, perception that older adults are at highest risk of COVID-19 and receiving COVID-19 related information from Radio/television and friends/family/neighbours.

**Funding:** The author(s) received no specific funding for this work.

**Competing interests:** The authors have declared that no competing interests exist.

## Conclusions

Our study highlighted that currently there little fear of COVID-19 among the older Rohingya FDMNs. This is probably due to lack of awareness of the severity of the disease in. Dissemination of public health information relevant to COVID-19 and provision of mental health services should be intensified particularly focusing on the individual who were concerned, overwhelmed or fearful of COVID-19. However, further qualitative research is advised to find out the reasons behind this.

## Introduction

The COVID-19 pandemic is one of the greatest calamities related to public health issue since World War II. The disease continue to spread across the world, with nearly 37 million confirmed cases in 188 countries with more than one million deaths up to 8[th] October 2020 [1]. World Health Organization (WHO) declared the disease as a global health emergency on 30 January 2020 [2]. All countries are currently implementing public health measures such as mobility restrictions, lockdowns, compulsory mask wearing, hand washing, among others to reduce the transmission of this highly infectious disease to control the morbidity as well as mortality rate.

However, public health emergencies in times of epidemics and pandemics such as the COVID-19 pandemic may cause intense fear and anxiety, particularly among the most at-risk populations including refugees and displaced population, particularly the old age group [3, 4]. This is because refugee camps are transitory settlements that accommodate displaced individuals and families who have fled from their home countries as a result of political instability, war and natural disasters [5]. In many refugee camps including the Cox's Bazar refugee camps of Bangladesh where Rohingya refuges are settled, there are poor social and physical conditions such as overcrowding, poor sanitation and absence of basic amenities such as water [5]. The poor social and physical conditions have negative implications for the adoption and adherence of safety and precautionary measures for COVID-19 management such as quarantine, hand washing, social distancing, among others [6].

Rohingya people (Forcibly Displace Myanmar Nationals or FDMN) are majorly Muslims minority groups who had been exiled from Rakhine State, Myanmar, for the fear of persecution and death. Estimates show that thousands of Rohingya people fled during August 2017 due to extreme violence. They took shelter in Cox's Bazar, a South-Eastern district of Bangladesh located about 280 kilometres away from Rakhine state of Myanmar (the place from where they displaced), is now also the world's largest camp for displaced people [7, 8]. Now, almost 860,000 FDMNs are living in the 34 camps in Cox's Bazar, with 51% being children, 45% adult, 4% older persons and 1% persons with disability [9].

The refugee camps are the most densely populated area where the average population density is about 40,000 inhabitants per square kilometre, with some areas approaching 70,000 inhabitants [8]. As a result, there is a high risk of transmission of COVID-19 in this area [10]. As of 30[th] September 2020, a total of 4479 COVID-19 positive cases have been confirmed in the Cox's Bazar district, of which 252 are in the Rohingya camps and the numbers are rapidly increasing [2]. People of all age groups are at risk of getting infected by this virus but older adults and people with pre-existing conditions are at higher risk of infection by COVID-19 [11]. The emerging evidence shows that mortality related to COVID-19 is higher (15%)

among the older adults worldwide, where almost 74% of the total death occurred among those above 65 years of age [12].

The UN Refugee Agency (UNHCR) indicates there are more than 31,500 FDMNs aged 60 years or older in the camps [13]. Therefore, age being a critical variable posing a major risk for COVID-19 may create serious emotional disturbances, insecurity, anxiety and depression among old, aged people in the camps. The effect of COVID-19 among the Rohingya older FDMNs might be devastating because of the prevalence of multiple health challenges in the refugee camp like lack of healthcare facilities and services, existing higher prevalence of infectious and non-communicable chronic diseases, and poor knowledge of hygiene and practices [14]. Again, a contagion disease outbreak like the COVID-19 creates fear that can cause people to worry about their own health and the health of the loved ones, financial situation, changes in sleep or eating patterns, difficulty in concentrating, worsening physical and mental health problems [11]. It is worth mentioning here that older people from the refugee camp in Rohingya community may have experiences of fear of military brutality which may exacerbate the situation of their mental/psychological condition mix with the fear of pandemic COVID-19. In this line, few media reports highlighted the COVID-19 fear among the Rohingya community of Cox's Bazar. However, there has been none scientific studies that measured the COVID-19 fear among this vulnerable group of population. Therefore, the current study was conducted to assess the COVID-19 related level of fear and its associates among FDMN older adults aged 60 years or above in Bangladesh.

## Materials and methods

### Study design and participants

This cross sectional study was carried out among 416 FDMNs older adults aged 60 years and above residing in Rohingya refugee camps situated in the Cox's Bazar district in the South-Eastern part of Bangladesh in October 2020.

The sample size of 460 was calculated with the following assumptions: (unknown) prevalence of COVID-19 related fear = 50%, sampling error = 5%, Confidence Interval = 95% and non-response rate = 20%. A total of 416 Rohingya older adults aged 60 years and above responded to the study from 457 who were approached (response rate 91%). There is a total of 34 Rohingya camps located in Cox's Bazar district, from which Camp 08E (SSID CXB-210), located at *Ukhia* sub-district was conveniently selected. In absence of the list of the older adults in Rohingya camps, a convenience sampling technique was employed to identify the eligible participants from the selected camp. The enumerators continued visiting the households starting from one side of the camp and stopped once desired sample size was achieved. In the absence of an eligible participant in the approached household, the enumerators moved to the next one. The inclusion criteria included aged ≥60 years and FDMN status. In absence of any registered document in most of the cases, this was confirmed by asking potential participants of their age and those who were 60 years and above were included. The exclusion criteria included adverse mental conditions (clinically proved schizophrenia, bipolar mood disorder, dementia/cognitive impairment), a hearing disability, or unable to communicate. We interviewed one eligible participant from each of the selected households. The oldest one was interviewed in case of more than one eligible participant in a selected household.

### Measures

**Outcome measure.** COVID-19 related fear was the primary outcome, which was measured using the seven-item Fear of COVID-19 Scale (FCV-19S) developed and validated by Ahorus et al. among the general Iranian population [15]. The FCV-19S is by far the most

widely used scale to measure the fear of COVID-19 and has previously been used among Bangladeshi people amid this COVID-19 pandemic [16, 17]. The reliability or the internal consistency of the scale among Rohingya older adults was also acceptable (Cronbach's α = 0.89).

Participants' agreement/disagreement with the seven items was assessed using a five-point Likert-scale (ranging from 1 = "strongly disagree", 2 = "disagree", 3 = "neither agree nor disagree", 4 = "agree", and 5 = "strongly agree"). Hence, the cumulative score ranged from 7 to 35, where the higher the scores, the greater the fear of COVID-19. We further classified the COVID-19 related fear into low fear (fear score below the mean of the scale value (7–35), i.e., <21) and high fear (fear score equal to or higher than the mean of the scale value (7–35), i.e., ≥21).

**Explanatory variables.**  Explanatory variables considered in this study were age (categorized as 60–69, 70–79, and ≥80 years), gender (male/female), marital status (currently married/widow), family size (≤4 and more than 4), literacy (Illiterate/literate), living arrangements (living with other family members/living alone), dependence on family for living (yes/no), memory or concentration problem (no problem/low memory or concentration), pre-existing non-communicable chronic conditions (yes/no), source of COVID-19 related information (Radio/Television, health workers, and friends/family/neighbors), concerned about COVID-19 (hardly, sometimes/often), overwhelmed by COVID-19 (hardly, sometimes/often), frequency of communication during COVID-19 (less than previous/same as previous), difficulty obtaining food during COVID-19 (no difficulty/faced difficulty), difficulty getting medicine during COVID-19 (no difficulty/faced difficulty), difficulty receiving routine medical care during COVID-19 (no difficulty/faced difficulty) and perceived that older adults at highest risk of COVID-19 (yes/no).

Self-reported information on the presence of pre-existing non-communicable conditions such as arthritis, hypertension, heart diseases, stroke, hypercholesterolemia, diabetes, chronic respiratory diseases, chronic kidney disease and cancer was also collected. This information was verified with health records of the participants (where these were available) and/or with family members. We did not collect information on the income and occupation as the participants were all unemployed and dependent on aid.

## Data collection tools and techniques

A pre-tested semi-structured questionnaire in Bengali language was used to collect the information through face-to-face interview. Data were electronically recorded in Survey CTO mobile app (https://www.surveycto.com/) by two surveyors, who were local residents of Cox's Bazar, fluent in Rohingya dialects, and had previous experience of administering health survey in electronic platform. The enumerators were trained extensively before the data collection through half-day Zoom meeting on the data collection tools and techniques as well as procedures of maintaining COVID-19 safe behaviors during the data collection.

The English version of the questionnaire was first translated to Bengali language and then back translated to English by two researchers to ensure the contents' consistency. The Bengali version of the tool was piloted among a small sample (n = 10) of Rohingya older adults from the selected camp to refine the language in the final version. The participants approved the tool translated in Bengali language by the research team without any corrections or modifications. Data collection was accomplished using this final tool through face-to-face interview of the participants and each of the interview took around half an hour.

## Statistical analysis

We performed descriptive analysis to assess the distribution of the variables. The level of fear (low/high fear score) was compared within different categories of a variable using chi square

test at 5% level of significance. A multiple linear regression model was performed where variable selection was based on the backward elimination with Akaike Information Criterion (AIC) approach. Adjusted beta-coefficient ($\beta$) and 95% confidence interval (95% CI) are reported. All analyses were performed using the statistical software package Stata (Version 14.0).

In case of the variable 'Source of COVID-19 related information' we prepared dummy variable for each of the reported source and fear score was compared for each dummy variable. For example, fear score was compared between the participants who reported that they received the COVID-19 related information radio/TV compared to those who did not receive the information from radio/TV.

### Ethical approval

The study protocol was approved by the institutional review board of Institute of Health Economics, University of Dhaka, Bangladesh (Ref: IHE/2020/1037). Both written and oral informed consents was sought from the participants (thumb impressions from those who were not able to read and write) before administering the survey. Participation was voluntary, and participants did not receive any compensation. Written approval was also sought from the Office of the Refugee Relief and Repatriation Commissioner (RRRC) prior to accessing the camps and conducting the survey. Participants also maintained COVID-19 safe behaviors during the interview such as practicing social distancing and wearing mask to protect the health of themselves and the participants.

### Patient and public involvement

Patients and/or public were not involved in development of research question, study design, conducting study and result dissemination.

## Results

### Participants' characteristics

Table 1 describes the socio-demographic characteristics of participants. Of 416 studied participants, 74% were aged 60–69 years, about 60% were male, 93% were married and almost 60% of them were residing with a household of more than four members. Nearly all the participants (98%) were functionally illiterate. A large number (87%) of the participants were living with the family members but half of them (46%) were not depending on their families for a living. Participants received information related to COVID-19 from different sources including Radio/Television (42.8%), health workers (72%) and friends/family/ neighbours (64%).

### Fear among the participants

Mean fear score was 14.8 (range 8–28) among the participants. Table 2 shows the bivariate analysis with participants' characteristics and categorized fear score of COVID-19. Overall, 88.9% of the participants had low fear score and 11.1% had high fear score. Here we also found that participants' age, gender, marital status, family size, living arrangement, having problem in memory and having pre-existing chronic conditions did not play a role in making them fearful of COVID-19. Fear score was significantly higher among the participants receiving COVID-19 information from family/friends, feeling concerned about and overwhelmed by COVID-19 and having same frequency of communication during COVID-19 compared to the previous ($P<0.001$). Also, the participants who were dependant on their family for living and who faced difficulty in obtaining food during COVID-19 had high fear score ($P<0.05$).

**Table 1. Sociodemographic characteristics of the participants (N = 416).**

| Characteristics | n | % |
|---|---|---|
| Total | 416 | 100.0 |
| Age (year, %) | | |
| 60–69 | 308 | 74.0 |
| 70–79 | 83 | 20.0 |
| > = 80 | 25 | 6.0 |
| Sex | | |
| Male | 251 | 60.3 |
| Female | 165 | 39.7 |
| Marital status | | |
| Married | 389 | 93.5 |
| Widow | 27 | 6.5 |
| Family size | | |
| 0–4 | 167 | 40.1 |
| >4 | 249 | 59.9 |
| Literacy | | |
| Illiterate | 406 | 97.6 |
| Literate | 10 | 2.4 |
| Living arrangement | | |
| Living with other family members | 362 | 87.0 |
| Living alone | 54 | 13.0 |
| Dependent on family for living | | |
| No | 191 | 45.9 |
| Yes | 225 | 54.1 |
| *Source of COVID-19 related information | | |
| Radio/Television | 178 | 42.8 |
| Health worker | 298 | 71.6 |
| Friends/family/neighbour | 267 | 64.2 |

## Independent factors associated with fear score

The full model included the socio-demographic characteristics and COVID-19 related variables (Table 1) deemed to be associated with COVID-19 related fear. The final model based on the lowest AIC are presented in Table 3. In the adjusted model, Radio/Television as source of COVID-19 related information, friends/family/neighbour as source of COVID-19 related information, overwhelmed by COVID-19, difficulty obtaining food during COVID-19 and frequency of communication during COVID-19 were significantly associated with COVID-19 related fear among study participants (Table 3).

COVID-19 related fear was 1.07 units higher among participants who received COVID-19 related information from Radio/Television ($\beta$: 1.07, 95% CI: 0.19 to 1.95) and 2.62 units higher among those who received COVID-19 related information from friends/family/neighbour ($\beta$: 2.62, 95% CI: 1.44 to 3.79). Similarly, the fear scores were 3.54 units higher among those who were overwhelmed by COVID-19 ($\beta$: 3.54, 95% CI: 2.54 to 4.55). On the other hand, the fear scores were 4.41 units lower among participants who had less communication with others during COVID-19 than previous ($\beta$: -4.41, 95% CI: -5.69 to -3.13). Meanwhile, the fear scores were 2.16 units higher among participants who had difficulty in obtaining food during COVID-19 ($\beta$: 2.16, 95% CI: 1.08 to -3.23).

**Table 2. Fear score and bivariate analysis (N = 416).**

| Characteristics | COVID-19 fear score | | |
|---|---|---|---|
| | Low (%) | High (%) | *P* |
| Overall | 88.9 | 11.1 | |
| Age (year, %) | | | |
| 60–69 | 87.3 | 12.7 | 0.204 |
| 70–79 | 94.0 | 6.0 | |
| > = 80 | 92.0 | 8.0 | |
| Gender | | | |
| Male | 89.6 | 10.4 | 0.575 |
| Female | 87.9 | 12.1 | |
| Marital status | | | |
| Married | 88.2 | 11.8 | 0.058 |
| Single | 100.0 | 0.0 | |
| Family size | | | |
| 0–4 | 92.2 | 7.8 | 0.081 |
| >4 | 86.8 | 13.3 | |
| Living arrangement | | | |
| Living with other family members | 88.1 | 11.9 | 0.167 |
| Living alone | 94.4 | 5.6 | |
| Dependent on family for living | | | |
| No | 92.7 | 7.3 | 0.025 |
| Yes | 85.8 | 14.2 | |
| Memory or concentration problems | | | |
| No problem | 88.8 | 11.2 | 0.778 |
| Low memory or concentration | 90.0 | 10.0 | |
| Pre-existing chronic conditions | | | |
| No | 87.5 | 12.5 | 0.132 |
| Yes | 92.6 | 7.4 | |
| *Source of COVID-19 related information | | | |
| Radio/Television | 95.5 | 4.5 | <0.001 |
| Health worker | 87.3 | 12.8 | 0.080 |
| Friends/family/neighbour | 83.9 | 16.1 | <0.001 |
| Concerned about COVID-19 | | | |
| Hardly | 98.1 | 2.0 | <0.001 |
| Sometimes/often | 74.4 | 25.6 | |
| Overwhelmed by COVID-19 | | | |
| Hardly | 97.0 | 3.0 | <0.001 |
| Sometimes/often | 80.9 | 19.1 | |
| Feeling of loneliness | | | |
| Hardly | 86.3 | 13.7 | 0.006 |
| Sometimes/often | 95.7 | 4.3 | |
| Frequency of communication during COVID-19 | | | |
| Same as previous | 75.9 | 24.1 | <0.001 |
| Less than previous | 98.0 | 2.0 | |
| Difficulty obtaining food during COVID-19 | | | |
| No difficulty | 89.4 | 10.6 | 0.292 |
| Faced difficulty | 85.7 | 14.3 | |
| Difficulty getting medicine during COVID-19 | | | |

(*Continued*)

**Table 2.** (Continued)

| Characteristics | COVID-19 fear score | | |
|---|---|---|---|
| | Low (%) | High (%) | *P* |
| No difficulty | 86.6 | 13.4 | 0.053 |
| Faced difficulty | 93.3 | 6.7 | |
| Difficulty receiving routine medical care during COVID-19 | | | |
| No difficulty | 86.2 | 13.8 | 0.047 |
| Faced difficulty | 93.2 | 6.8 | |
| Perceived that older adults at highest risk of COVID-19 | | | |
| No | 89.4 | 10.6 | 0.821 |
| Yes | 88.7 | 11.3 | |

*Multiple responses.

## Discussion

To our knowledge, this is the first study to assess the level of fear and its associates among Rohingya FDMNs older adults in Cox's Bazar refugee camp. This study revealed two major findings. First, there was a relatively low level of fear among participants. Secondly, fear was associated with some socio-demographic characteristics of the participants and COVID-19 related factors.

   Our study provides evidence to contradict the general assumption that COVID-19 has created enormous fear among all population groups, especially among the aged population in the world [18–20]. Although our findings are somewhat different with previous studies, it is worth

**Table 3. Factors associated with fear among the participants (N = 416).**

| Characteristics | *β* | *P* | 95% CI |
|---|---|---|---|
| Radio/Television as source of COVID-19 related information | | | |
| No | *Ref* | | |
| Yes | 1.07 | 0.017 | 0.19, 1.95 |
| Friends/family/neighbour as source of COVID-19 related information | | | |
| No | *Ref* | | |
| Yes | 2.62 | <0.001 | 1.44, 3.79 |
| Concerned about COVID-19 | | | |
| Hardly | *Ref* | | |
| Sometimes/often | 0.63 | 0.166 | -0.26, 1.53 |
| Overwhelmed by COVID-19 | | | |
| Hardly | *Ref* | | |
| Sometimes/often | 3.54 | <0.001 | 2.54, 4.55 |
| Frequency of communication during COVID-19 | | | |
| Same as previous | *Ref* | | |
| Less than previous | -4.41 | <0.001 | -5.69, -3.13 |
| Difficulty obtaining food during COVID-19 | | | |
| No difficulty | *Ref* | | |
| Faced difficulty | 2.16 | <0.001 | 1.08, 3.23 |
| Perceived that older adults at highest risk of COVID-19 | | | |
| No | *Ref* | | |
| Yes | 1.26 | 0.064 | -0.07, 2.59 |

mentioning that the included studies were not only performed with refugees in camps, but also used different measures and scales. Fear remains one of the key causes of stress, anxiety, and mental problems (WHO, 2020a). Thus, our participants having low level of fear of COVID-19 can be considered a positive condition for their mental and psychological wellbeing. However, this finding is particularly interesting and needs some explanations considering evidence that older adults are more likely to be infected and die from COVID-19 [21–24]. There can be several reasons for less fear for COVID-19 among the older Rohingya participants, such as poor health literacy about this infectious disease, previous brutal experience back in Myanmar, low COVID-19 cases in camp, low exposure to media in the camps and so on.

The present study further noted the factors associated with the higher level of fear among the older Rohingya population. We found that the participants who frequently become overwhelmed by the lethality of COVID-19 were more fearful than those who were indifferent to it. This is expected as people often become anxious and stressful of a disease when they are overwhelmed by its deadly effect, which in turn make them fearful of it. Evidence suggests that older adults are more vulnerable to COVID-19 related death and disabilities than their younger counterpart in part due to the presence of several co-morbidities [25]. Thus, knowing that presently there is no cure for COVID-19 and that they are more vulnerable of facing its lethal outcomes, they might have been fearful of being infected with it [26, 27]. In this light, local health authorities and international partners should develop infrastructure to provide services for pre-existing conditions and testing services for COVID 19 along with other promotive and preventive services, such as adequate supply of masks, soaps, and sanitizers during the COVID-19 epidemic.

Interestingly, the findings revealed that the participants who had relatively lesser communication with others during the pandemic than before was less fearful than those who had no effect with communication. This is probably because frequent discussion of the severity of COVID-19, its global reach and the number of deaths and disabilities it is causing around them and globally with friends, family and other acquaintances can make a person more fearful of it [28]. Our findings also support this, as we found that the participants who were receiving COVID-19 related information from their family/friends/neighbours were more fearful than those who were not getting information from them. In line with this, the present research also reports that receiving COVID-19 information from popular media sources i.e., Radio/ Television also made the participants fearful of it. Again, the similar explanation applies as the participants are getting several information related to the lethality of the disease through Radio/Television making them fearful of it. Moreover, it is obvious that at the time of any pandemic overwhelming number of popular media sources pick up fearful news about impact of the virus. In this line, it is crucial to encourage a more positive approach while delivering COVID-19 information alongside this challenging time. We also suggest the need to provide correct COVID-19 information in diagrammatic format designed in plain understandable languages to support those with no formal education.

We also found that the participants who had difficulty in obtaining food during this pandemic were more fearful than those who had not difficulty. Like many other countries of the world, earning members in many of the families lost their job in Bangladesh during the lockdown imposed due to COVID-19, particularly among the production workforce [29]. Also, crops and vegetables were not harvested in time or couldn't transport to the market in time which caused food shortage and hiked off the price [30]. Peoples' earning are reducing day by day due to the disease outbreak on one side, on the other hand, the price of the foods are increasing playing a significant role making them fearful of deadly effect of the disease.

Moreover, we found that the participants who believed that older adults are at highest risk of COVID-19 were more fearful. The possible explanation could be that the participants who

had a feeling that they are at highest risk of being infected by COVID-19 and infection with COVID-19 could be very lethal may have greater level of fear. Moreover, participants with such feeling would have been very anxious of being infected with the diseases and remain fearful of it [31].

To the best of our knowledge, this is the first study carried out among Rohingya older adults providing an insight of fear amid this COVID-19 pandemic. However, our study has some limitations too. First, we prepared our sampling frame based on the available household-level information in our data repository; thus, selection bias is possible. Second, recall bias and social desirability could be potentially inevitable in this study as all items and measures used were self-reported. Thirdly, the relationship between fear and socio-demographic and lifestyle factors can be complex, however, our analysis was based on a cross-sectional survey data, which may not be able to establish causation links. Also, our data prohibit making any causal and directional conclusions since we could not eliminate the potential effect due to reverse causality. Fourth, in the absence of pre-pandemic estimates, we cannot assert that the increased prevalence of fear noted in our study could be attributed to the COVID-19 pandemic. Also, the tool used to measure the level of fear was not developed with refugees. These and other limitations may affect the veracity of the findings that may limit the representation and generalizability of the findings.

## Conclusion

Low fear of COVID-19 pandemic among the older refugees in the Rohingya camp suggests that they are less concerned about the devastating appearance of the current pandemic. This may be due not to apprehending the actual situation, illiteracy/lack of knowledge about this infectious disease, previous brutal experience, and so forth. Fearlessness, though good in a sense of not having mental pressure among the most vulnerable peoples, could exacerbate the situation in the camp all in a sudden as they won't take any precautionary steps to combat the virus infection. Dissemination of the actual information, withdrawn the restriction on accessing media/ internet, availability of the health care facilities including the test, facilities will improve knowledge among the older Rohingya in the refugee camp will prevent further deterioration of the condition due to this COVID-19. Further qualitative study should have to be conducted to depict the real picture or the reasons behind the fearless situation of the older Rohingya FDMNs.

## Supporting information

**S1 File. Data file of the study.**
(DTA)

**S2 File. English questionnaire.**
(PDF)

**S3 File. Bengali questionnaire.**
(PDF)

## Acknowledgments

We acknowledge the role of Sadia Sumaia Chowdhury, Programme Manager, ARCED Foundation and Muntasir Alam, Research Assistant, ARCED Foundation for their support in data collection for the study.

## Author Contributions

**Conceptualization:** Sabuj Kanti Mistry, A. R. M. Mehrab Ali, Prince Peprah, Uday Narayan Yadav.

**Data curation:** A. R. M. Mehrab Ali.

**Formal analysis:** Sabuj Kanti Mistry.

**Investigation:** Sabuj Kanti Mistry, A. R. M. Mehrab Ali.

**Methodology:** Sabuj Kanti Mistry, A. R. M. Mehrab Ali, Uday Narayan Yadav.

**Project administration:** Sabuj Kanti Mistry, A. R. M. Mehrab Ali, Sompa Reza.

**Software:** Sabuj Kanti Mistry.

**Supervision:** A. R. M. Mehrab Ali, Sompa Reza, Uday Narayan Yadav.

**Validation:** Sabuj Kanti Mistry.

**Visualization:** Sabuj Kanti Mistry.

**Writing – original draft:** Sabuj Kanti Mistry, A. R. M. Mehrab Ali, Farhana Akther, Sompa Reza, Shaidatonnisha Prova, Uday Narayan Yadav.

**Writing – review & editing:** Uday Narayan Yadav.

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
