## [Decision Letter · Decision Letter 0]

26 Apr 2021

PONE-D-21-00805

Do older adults of Rohingya community (Forcibly Displaced Myanmar Nationals or FDMNs) in Bangladesh are fearful of COVID-19? findings from a cross-sectional study

PLOS ONE

Dear Author,

Thank you for submitting your manuscript to PLOS ONE. After careful consideration, we feel that it has merit but does not fully meet PLOS ONE’s publication criteria as it currently stands. Therefore, we invite you to submit a revised version of the manuscript that addresses the points raised during the review process.

We look forward to receiving your revised manuscript.

Kind regards,

Ramesh Kumar, PhD

Academic Editor

PLOS ONE

Journal Requirements:

2. Please note that PLOS ONE does not copy edit accepted manuscripts (https://journals.plos.org/plosone/s/criteria-for-publication#loc-5). To that effect, please ensure that your submission is free of typos and grammatical errors, including the title.

4. We note that Figure 1 in your submission contain map images which may be copyrighted. All PLOS content is published under the Creative Commons Attribution License (CC BY 4.0), which means that the manuscript, images, and Supporting Information files will be freely available online, and any third party is permitted to access, download, copy, distribute, and use these materials in any way, even commercially, with proper attribution. For these reasons, we cannot publish previously copyrighted maps or satellite images created using proprietary data, such as Google software (Google Maps, Street View, and Earth). For more information, see our copyright guidelines: http://journals.plos.org/plosone/s/licenses-and-copyright.

4.1.    You may seek permission from the original copyright holder of Figure 1 to publish the content specifically under the CC BY 4.0 license. 

4.2.    If you are unable to obtain permission from the original copyright holder to publish these figures under the CC BY 4.0 license or if the copyright holder’s requirements are incompatible with the CC BY 4.0 license, please either i) remove the figure or ii) supply a replacement figure that complies with the CC BY 4.0 license. Please check copyright information on all replacement figures and update the figure caption with source information. If applicable, please specify in the figure caption text when a figure is similar but not identical to the original image and is therefore for illustrative purposes only.

Reviewers' comments:

Reviewer's Responses to Questions

**Comments to the Author**

1. Is the manuscript technically sound, and do the data support the conclusions?

Reviewer #1: Yes

Reviewer #2: Yes

2. Has the statistical analysis been performed appropriately and rigorously? 

Reviewer #1: Yes

Reviewer #2: Yes

3. Have the authors made all data underlying the findings in their manuscript fully available?

Reviewer #1: Yes

Reviewer #2: Yes

4. Is the manuscript presented in an intelligible fashion and written in standard English?

Reviewer #1: Yes

Reviewer #2: No

5. Review Comments to the Author

Reviewer #1: Reviewer comment

Abstract:

In conclusion highlighted part is not reflected in result.

Introduction :

First paragraph: In highlighted part please add hand washing issue as a preventive measure.

Materials and Methods:

Please mention how age in inclusion criteria was confirmed?

Outcome measures:

To put operational definition of higher / lower score of fear

Reviewer #2: This paper contributes a very important and timely piece of work to the discussion around the impacts of COVID-19 on unreached and often disadvantaged populations. The authors should be commended on their ability to collect data from such a large sample size given the challenging circumstances associated with COVID-19.

Prior to acceptance of this research for publication, the following areas should be addressed:

1. This manuscript would benefit from extensive English language editing to improve readability.

Abstract:

2. In the abstract the authors note that the mean fear score was 14.8, but give no context as to what this score means. Authors should consider including the maximum possible fear score, ie ‘a mean score of 14.8 out of a possible XX’.

3. Additionally, it seems odd as a reader that the conclusion does not refer to strategies for individuals who were concerned, overwhelmed or fearful of COVID, given the reference to this group in the results.

Methods

4. If there were more than one person from a household interviewed, were interviews done separately or together. Were multiple individuals from the same household treated differently in the analysis (ie were there any tests done to determine whether this skewed results?)

5. How long did the survey take to complete (on average)?

6. Following the piloting of the survey in Bengali, were there any changes/modification to the original survey? Or to suit the specific cohort?

7. Justification is needed of why this scale was used (discussion refers to other available scales).

8. Did the survey collect information on covid-safe behaviours (ie handwashing, ability to practice social distance), presumedly these would have an impact on fear also or lack of fear may impact on not engaging in behaviours?

9. For the ethics section, how did the researchers maintain COVID-safe practices to protect the health of themselves and the participants. Were there any additional approvals required relating to this?

10. Were there any actions to mitigate the potential influence of perceived power (from the data collectors)? Ie did researchers have a previous relationship with participants? Given the stated concern of the community with authority/military, how did the researchers mitigate potential power imbalances? This includes both in participants giving consent (were there any measures beyond being told participation was voluntary?) and during interviews (how did researchers ensure participants did not give the responses they felt were wanted)?

Results

11. Given that the authors have only provided an analysis of mean scores, were there some individuals that reported ‘high’ scores? If so, what would be coded as high and how many fell into this category?

12. Table 2: Why does “Source of COVID-19 related information” have a *? There are no notes at the bottom of the Table

13. What has the p value for “Source of COVID-19 related information” been compared with? The other categories seem to compare mean scores within each category, has this score been compared to those who did not select those sources? If so, please include in the methods.

14. For Concerned/ overwhelmed about COVID-19 categories, was there a none option? Or were the options only ‘hardly’ or ‘sometimes/often’?

Discussion

15. Please clarify why there is a fairly low level of fear. Is this because the mean scores (around 15 out of a possible 35) have been categorised as low? Is this compared to the scores collected in the original COVID fear study?

16. The paper seems to have two limitations sections, these could be merged.

The following provides some suggestions the authors may choose to include to strengthen the manuscript:

Abstract

1. In the conclusion section the authors state that there was very little fear among the study sample. Given this point, it may be better to include the % of the sample who has a low fear score (instead of using the mean fear score as currently in the document).

2. Where the authors state “lack of awareness of the severity of the disease and brutality the older Rohingya adults faced in recent past which OUTRAGED this fear”, do you mean increased or exacerbated this fear, instead of outraged?

Introduction

3. Please include the year on the end of the following text: with more than one million deaths up to 8th October

4. When referring to the “poor social and physical conditions” in refugee camps, authors may wish to consider reflecting on the impact this has on COVID transmission. Ie the inability to quarantine cases, lack of WASH to reduce spread, lack of PPE.

5. Please ensure consistent terminology throughout the text, eg sometimes COVID, sometimes COVID-19

Results

6. The authors should consider including detail on the range of fear scores collected (rather than only reporting mean scores)-perhaps in a frequency table?

Discussion

7. Please clarify why there is a fairly low level of fear. Is this because the mean scores (around 15 out of a possible 35) have been categorised as low? Is this compared to the scores collected in the original COVID fear study?

8. The paper seems to have two limitations sections, these could be merged.

9. While the discussion mentions the “frequent discussion of the severity of COVID-19.”as impacting on fear, I wonder if the authors have considered the need to counter these messages and/or whether there is scope to include this in their manuscript? The WHO has developed resources to address stigma associated with COVID-19, such as https://www.who.int/docs/default-source/coronaviruse/covid19-stigma-guide.pdf.

Conclusion

10. The conclusion makes some excellent that the authors could expand further in the discussion if they have room to do so. For example, Dissemination of the actual information, availability of the health care facilities and further qualitative study

6. PLOS authors have the option to publish the peer review history of their article (what does this mean?). If published, this will include your full peer review and any attached files.

Reviewer #1: No

Reviewer #2: No

---

## [Author Response · Author response to Decision Letter 0]

13 May 2021

Re: "Are older adults of Rohingya community (Forcibly Displaced Myanmar Nationals or FDMNs) in Bangladesh fearful of COVID-19? findings from a cross-sectional study"

Dear Editor,

We greatly appreciate the valuable comments from the Editor and the reviewers and have modified the enclosed manuscript accordingly. Here, we include our responses to each of the reviewers’ comments.

Editor

https://journals.plos.org/plosone /s/file?id=wjVg/PLOSOne_formatting_

sample_main_body.pdf and

Review response: We confirm that the manuscript complies with the PLOS One guidelines

2. Please note that PLOS ONE does not copy edit accepted manuscripts (https://journals.plos.org/plosone/s/criteria-for-publication#loc-5). To that effect, please ensure that your submission is free of typos and grammatical errors, including the title.

Review response: We ensure that the manuscript is free of typos and grammatical erros.

We have included the questionnaire in both English and Bengali (Please see supplementary file S2 and S3)

4. We note that Figure 1 in your submission contain map images which may be copyrighted. All PLOS content is published under the Creative Commons Attribution License (CC BY 4.0), which means that the manuscript, images, and Supporting Information files will be freely available online, and any third party is permitted to access, download, copy, distribute, and use these materials in any way, even commercially, with proper attribution. For these reasons, we cannot publish previously copyrighted maps or satellite images created using proprietary data, such as Google software (Google Maps, Street View, and Earth). For more information, see our copyright guidelines: http://journals.plos.org/plosone/s/licenses-and-copyright.

4.1. You may seek permission from the original copyright holder of Figure 1 to publish the content specifically under the CC BY 4.0 license. 

4.2. If you are unable to obtain permission from the original copyright holder to publish these figures under the CC BY 4.0 license or if the copyright holder’s requirements are incompatible with the CC BY 4.0 license, please either i) remove the figure or ii) supply a replacement figure that complies with the CC BY 4.0 license. Please check copyright information on all replacement figures and update the figure caption with source information. If applicable, please specify in the figure caption text when a figure is similar but not identical to the original image and is therefore for illustrative purposes only.

Review response: We have removed the figure.

Reviewer 1

Abstract:

In conclusion highlighted part is not reflected in result.

Review response: Thanks to the reviewer for bringing this to our notice. As agree, we have modified the Abstract to reflect the results of the study.

Introduction:

First paragraph: In highlighted part please add hand washing issue as a preventive measure.

Review response: Thanks, hand washing as a preventative measure has been added. Please see page 4 line 62.

Materials and Methods:

Please mention how age in inclusion criteria was confirmed?

Review response: Thanks again. A statement on how the age criterion was confirmed has been provided in the revised manuscript. We have included “In absence of any registered document in most of the cases, this was confirmed by asking potential participants of their age and those who were 60 years and above were included.” Please see page 7 line 130-132.

Outcome measures:

To put operational definition of higher / lower score of fear

Review response: Now, we have provided the operational definition of low/high fear score. Please see page 8 line 152-154.

Reviewer 2

This paper contributes a very important and timely piece of work to the discussion around the impacts of COVID-19 on unreached and often disadvantaged populations. The authors should be commended on their ability to collect data from such a large sample size given the challenging circumstances associated with COVID-19.

Prior to acceptance of this research for publication, the following areas should be addressed:

This manuscript would benefit from extensive English language editing to improve readability.

Review response: We have thoroughly checked the manuscript for English language and improving readability.

Abstract:

1. In the abstract the authors note that the mean fear score was 14.8, but give no context as to what this score means. Authors should consider including the maximum possible fear score, ie ‘a mean score of 14.8 out of a possible XX’.

Review response: Thanks very much for noting this. We have now mentioned in the Method of the Abstract that the cumulative fear score ranged from 7-35. Please see page 2 line 32-33. We also provided the range of fear score among the participants as well as the percentage of people with low fear in the Result of the Abstract. Please see page 2 line 35-36.

2. Additionally, it seems odd as a reader that the conclusion does not refer to strategies for individuals who were concerned, overwhelmed or fearful of COVID, given the reference to this group in the results.

Review response: Thanks to the reviewer for this important comment. Accordingly, we have revised this section. We have offered some strategies for helping individuals who were fearful of COVID-19. Please see page 2 line 47-50.

3. In the conclusion section the authors state that there was very little fear among the study sample. Given this point, it may be better to include the % of the sample who has a low fear score (instead of using the mean fear score as currently in the document).

Review response: Now we have added the percentage of people with low fear score.

4. Where the authors state “lack of awareness of the severity of the disease and brutality the older Rohingya adults faced in recent past which OUTRAGED this fear”, do you mean increased or exacerbated this fear, instead of outraged?

Review response: Thanks to the reviewer. As the conclusion part of the abstract has been revised this particular point has been changed. Please see page 2 line 45-50.

Introduction:

1. Please include the year on the end of the following text: with more than one million deaths up to 8th October

Review response: The year (2020) has been included. Please see page 4 line 59.

2. When referring to the “poor social and physical conditions” in refugee camps, authors may wish to consider reflecting on the impact this has on COVID transmission. Ie the inability to quarantine cases, lack of WASH to reduce spread, lack of PPE.

Review response: Thanks for this suggestion. We have accordingly included the implication that these social and physical conditions have for effective safety and precautionary measures implementation Please see page 4 line 72-74.

3. Please ensure consistent terminology throughout the text, eg sometimes COVID, sometimes COVID-19

Review response: Thanks for this observation. We have consistently used ‘COVID-19’ throughout the revised manuscript.

Methods:

1. If there were more than one person from a household interviewed, were interviews done separately or together. Were multiple individuals from the same household treated differently in the analysis (ie were there any tests done to determine whether this skewed results?)

Review response: Thanks very much for pointing this. We interviewed one eligible participant from each of the selected households. The oldest one was interviewed in case of more than one eligible participant in a selected household. We have added this in the Method section (page 7 line 134-136).

2. How long did the survey take to complete (on average)?

Review response: Thanks for pointing this. The survey took around an average of 30 minutes. We have added this in the revised manuscript. Please see page 9 line 193-194.

3. Following the piloting of the survey in Bengali, were there any changes/modification to the original survey? Or to suit the specific cohort?

Review response: As we have mentioned, the participants approved the tool translated in Bengali language by the research team without any corrections or modifications. Please see page 9 line 191-193.

4. Justification is needed of why this scale was used (discussion refers to other available scales).

Review response: We have added the following line: “The FCV-19S is by far the most widely used scale to measure the fear of COVID-19 and has previously been used among Bangladeshi people amid this COVID-19 pandemic.” Please see page line 7 line 144-146.

5. Did the survey collect information on covid-safe behaviours (ie handwashing, ability to practice social distance), presumedly these would have an impact on fear also or lack of fear may impact on not engaging in behaviours?

Review response: Yes, but we humbly wish to state that this was not the focus of the present manuscript. The impact of these Covid-19 safe behaviours on issues such as fear and mental health would likely form a separate paper on its own. However, thanks for this comment.

6. For the ethics section, how did the researchers maintain COVID-safe practices to protect the health of themselves and the participants. Were there any additional approvals required relating to this?

Review response: Participants were instructed to maintain COVID-19 safe behaviours during the interview such as practicing social distancing and wearing mask to protect the health of themselves and the participants. We have added this in the revised manuscript. Please see page 9 line 185-186 and page 11 line 219-221.

As we mentioned we received the ethical approval from Institute of Health Economics, University of Dhaka, Bangladesh (Ref: IHE/2020/1037) and the Office of the Refugee Relief and Repatriation Commissioner (RRRC) and we mentioned in the ethics application that we would follow COVID-19 safe practices while collecting the data.

7. Were there any actions to mitigate the potential influence of perceived power (from the data collectors)? Ie did researchers have a previous relationship with participants? Given the stated concern of the community with authority/military, how did the researchers mitigate potential power imbalances? This includes both in participants giving consent (were there any measures beyond being told participation was voluntary?) and during interviews (how did researchers ensure participants did not give the responses they felt were wanted)?

Review response: We did not apply any power imbalance strategy as you have mentioned. However, we would like to mention that none of the researchers have previous relationship with the participants, and the data collectors were also selected from outside the camp, and they were not familiar with the participants.

Results:

1. Given that the authors have only provided an analysis of mean scores, were there some individuals that reported ‘high’ scores? If so, what would be coded as high and how many fell into this category?

Review response: We have reanalyzed the data based on your comment and we have revised table 2 and relevant text. As you can see only 11.1% of the participants had high fear score. Please see page 12-13 line 241-261 of the revised manuscript and Table 2. 

2. Table 2: Why does “Source of COVID-19 related information” have a *? There are no notes at the bottom of the Table

Review response: Thanks for noting this. * denotes multiple responses. We have now added this at the bottom of the Table 2 as footnotes.

3. What has the p value for “Source of COVID-19 related information” been compared with? The other categories seem to compare mean scores within each category, has this score been compared to those who did not select those sources? If so, please include in the methods.

Review response: In case of the variable ‘Source of COVID-19 related information’ we prepared dummy variable for each of the reported source and fear score was compared for each dummy variable. For example, fear score was compared between the participants who reported that they received the COVID-19 related information radio/TV compared to those who did not receive the information from radio/TV. We have included this in the Method section (page 10 line 206-210)

4. For Concerned/ overwhelmed about COVID-19 categories, was there a none option? Or were the options only ‘hardly’ or ‘sometimes/often’?

Review response: We did not have ‘none’ option; instead the options were ‘hardly’ or ‘sometimes/often’.

5. The authors should consider including detail on the range of fear scores collected (rather than only reporting mean scores)-perhaps in a frequency table?

Review response: We have reanalyzed the data based on your comment and we have revised table 2 and relevant text. Please see page 12-13 line 241-261 of the revised manuscript and Table 2.

Discussion:

Please clarify why there is a fairly low level of fear. Is this because the mean scores (around 15 out of a possible 35) have been categorised as low? Is this compared to the scores collected in the original COVID fear study?

Review response: We have reanalyzed the data based on your comment with an operational definition of high/low fear score (page 8 line 152-154). As you can see in table 2, 88.9% of the participants had low fear score. 

1. The paper seems to have two limitations sections, these could be merged.

Review response: Thanks very much for this critical observation. We have accordingly merged the information. Please see page 19–20 line 361-381.

2. While the discussion mentions the “frequent discussion of the severity of COVID-19.”as impacting on fear, I wonder if the authors have considered the need to counter these messages and/or whether there is scope to include this in their manuscript? The WHO has developed resources to address stigma associated with COVID-19, such as https://www.who.int/docs/default-source/coronaviruse/covid19-stigma-guide.pdf.

Review response: Thanks much for your suggestion and we feel this is really something that we need to consider. Unfortunately, we have not measured COVID-19 related stigma in this work. We will consider this valuable point in our upcoming work.

Conclusion:

1. The conclusion makes some excellent that the authors could expand further in the discussion if they have room to do so. For example, Dissemination of the actual information, availability of the health care facilities and further qualitative study

Review response: Thanks much for your suggestion. We have revised the Discussion considering your valuable suggestions. Please see page 17-18 line 319-322 and page 18 line 336-340.

Concluding Remarks

We are extremely appreciative of the Editor’s and reviewers’ time and helpful comments, and hope that our revisions have adequately addressed their concerns. We are confident that the revisions have strengthened our manuscript. We look forward to hearing from you with a final decision regarding the acceptance of our manuscript.

Regards,

Authors

---

## [Decision Letter · Decision Letter 1]

10 Jun 2021

Are older adults of Rohingya community (Forcibly Displaced Myanmar Nationals or FDMNs) in Bangladesh fearful of COVID-19? findings from a cross-sectional study

PONE-D-21-00805R1

Dear Dr. Mistry,

We’re pleased to inform you that your manuscript has been judged scientifically suitable for publication and will be formally accepted for publication once it meets all outstanding technical requirements.

Kind regards,

Ramesh Kumar, PhD

Academic Editor

PLOS ONE

Additional Editor Comments (optional):

Reviewers' comments:

Reviewer's Responses to Questions

**Comments to the Author**

1. If the authors have adequately addressed your comments raised in a previous round of review and you feel that this manuscript is now acceptable for publication, you may indicate that here to bypass the “Comments to the Author” section, enter your conflict of interest statement in the “Confidential to Editor” section, and submit your "Accept" recommendation.

Reviewer #1: (No Response)

Reviewer #2: All comments have been addressed

2. Is the manuscript technically sound, and do the data support the conclusions?

Reviewer #1: (No Response)

Reviewer #2: Yes

3. Has the statistical analysis been performed appropriately and rigorously? 

Reviewer #1: (No Response)

Reviewer #2: Yes

4. Have the authors made all data underlying the findings in their manuscript fully available?

Reviewer #1: (No Response)

Reviewer #2: Yes

5. Is the manuscript presented in an intelligible fashion and written in standard English?

Reviewer #1: (No Response)

Reviewer #2: Yes

6. Review Comments to the Author

Reviewer #1: (No Response)

Reviewer #2: (No Response)

7. PLOS authors have the option to publish the peer review history of their article (what does this mean?). If published, this will include your full peer review and any attached files.

Reviewer #1: No

Reviewer #2: No

---

## [Editor Report · Acceptance letter]

14 Jun 2021

PONE-D-21-00805R1 

Are older adults of Rohingya community (Forcibly Displaced Myanmar Nationals or FDMNs) in Bangladesh fearful of COVID-19? findings from a cross-sectional study 

Dear Dr. Mistry:

I'm pleased to inform you that your manuscript has been deemed suitable for publication in PLOS ONE. Congratulations! Your manuscript is now with our production department. 

Kind regards, 

on behalf of

Dr. Ramesh Kumar 

Academic Editor

PLOS ONE